# Combinatorial expression of *ebony* and *tan* generates body color variation from nymph through adult stages in the cricket, *Gryllus bimaculatus*

**Shintaro Inoue[1], Takahito Watanabe[1], Taiki Hamaguchi[2], Yoshiyasu Ishimaru[3], Katsuyuki Miyawaki[1], Takeshi Nikawa[4], Akira Takahashi[5], Sumihare Noji[1], Taro Mito[1]***

**1** Bio-Innovation Research Center, Tokushima University, Ishii, Ishii-cho, Myozai-gun, Tokushima, Japan,
**2** Division of Bioresource Science, Graduate School of Sciences and Technology for Innovation, Tokushima University, Minami-Jyosanjima-cho, Tokushima, Japan, **3** Division of Bioscience and Bioindustry, Graduate School of Technology, Industrial and Social Sciences, Tokushima University, Minami-Jyosanjima-cho, Tokushima, Japan, **4** Departments of Nutritional Physiology, Institute of Biomedical Sciences, Tokushima University Graduate School, Kuramoto-cho, Tokushima, Japan, **5** Department of Preventive Environment and Nutrition, Institute of Biomedical Sciences, Tokushima University Graduate School, Kuramoto-cho, Tokushima, Japan

* mito.taro@tokushima-u.ac.jp

**Data Availability Statement:** The sequences of Gb'ebony and Gb'tan have been deposited into the DNA data bank of Japan under accession no. LC733201(Gb'ebony) and LC733202 (Gb'tan).

## Abstract

Insect body colors and patterns change markedly during development in some species as they adapt to their surroundings. The contribution of melanin and sclerotin pigments, both of which are synthesized from dopamine, to cuticle tanning has been well studied. Nevertheless, little is known about how insects alter their body color patterns. To investigate this mechanism, the cricket *Gryllus bimaculatus*, whose body color patterns change during postembryonic development, was used as a model in this study. We focused on the *ebony* and *tan* genes, which encode enzymes that catalyze the synthesis and degradation, respectively, of the precursor of yellow sclerotin *N*-β-alanyl dopamine (NBAD). Expression of the *G. bimaculatus* (*Gb*) *ebony* and *tan* transcripts tended to be elevated just after hatching and the molting period. We found that dynamic alterations in the combined expression levels of *Gb'ebony* and *Gb'tan* correlated with the body color transition from the nymphal stages to the adult. The body color of *Gb'ebony* knockout mutants generated by CRISPR/Cas9 systemically darkened. Meanwhile, *Gb'tan* knockout mutants displayed a yellow color in certain areas and stages. The phenotypes of the *Gb'ebony* and *Gb'tan* mutants probably result from an over-production of melanin and yellow sclerotin NBAD, respectively. Overall, stage-specific body color patterns in the postembryonic stages of the cricket are governed by the combinatorial expression of *Gb'ebony* and *Gb'tan*. Our findings provide insights into the mechanism by which insects evolve adaptive body coloration at each developmental stage.

Other relevant data are within the paper and its Supporting Information files.

**Funding:** SI, TW, KM, TN, AT, TM were supported by the Cabinet Office, Government of Japan, Cross-ministerial Moonshot Agriculture, Forestry and Fisheries Research and Development Program, "Technologies for Smart Bio-industry and Agriculture" (BRAIN) (JPJ009237). The funders had no role in study design, data collection and analysis, decision to publish, or preparation of the manuscript.

**Competing interests:** The authors have declared that no competing interests exist.

**Abbreviations:** CRISPR, clustered regularly interspaced short palindromic repeat; Cas9, CRISPR-associated proteins 9; NBAD, *N*-β-alanyl dopamine; NADA, *N*-acetyl dopamine; gRNA, guide RNA; crRNA, CRISPR RNA; DOPA, L-3,4-dihydroxyphenylalanine..

## Introduction

Body color is one of the most diversified features in insect morphology and is involved in camouflage, aposematism, and other processes [1, 2]. In addition to the evolutional diversity of body color patterns observed between and within species such as flies [2], it changes throughout the developmental stages of insects. Nymphs and adults of *Hymenopus coronatus* (known as the Orchid Mantis) show distinct color patterns. The first instar nymphs show a black-red color pattern, the second to last instar nymphs show a flower-like pattern, and the adults show a brown-white color pattern [3], and these color patterns are adaptive to the surrounding environment.

The cricket *Gryllus bimaculatus* (common name, two-spotted cricket) is a model hemimetabolous insect species [4]. Crickets, especially *G. bimaculatus*, have recently gained attention as a biological material for protein production [5, 6]. Studying gene functions in this insect is advantageous because of the availability of genomic data [7] and genome-editing technologies, including clustered regularly interspaced short palindromic repeats (CRISPR)/CRISPR-associated proteins 9 (Cas9) [8, 9]. This cricket species exhibits stage-specific body color patterns of a black-yellow-brown combination during postembryonic development; thus, it is suitable for analyzing the molecular mechanisms of the body color transition between developmental stages.

During the pigmentation process, the exoskeleton cuticle acquires coloration through a combination of melanin and sclerotin pigments. Previous studies in insects such as *Drosophila melanogaster* [2] and *Tribolium casteneum* [10] have demonstrated that dopamine is the core substrate in the model pathway of melanin and sclerotin biosynthesis. As shown in Fig 1, the biosynthesis of melanin and sclerotin starts from tyrosine, which is converted into L-3,4-dihydroxyphenylalanine (DOPA) by tyrosine hydroxylase (*pale*), the rate-limiting enzyme in this pathway. DOPA is converted into dopamine by DOPA decarboxylase (*ddc*), and dopamine is then used as a substrate to produce dopamine-melanin, a black pigment produced by the dopamine chrome conversion activity of the Yellow protein (*yellow*) and the phenol oxidase activity of Laccase2 (*lac2*). Excess dopamine is metabolized into *N*-β-alanyl dopamine (NBAD) and *N*-acetyl dopamine (NADA) by NBAD synthase (*ebony*) and dopamine *N*-acetyltransferase (*nat*), respectively. NBAD and NADA are also metabolized by Laccase2 to synthesize yellow NBAD sclerotin and colorless NADA sclerotin, respectively. In addition to these enzymes, this pathway may include NBAD hydroxylase (*tan*), which degrades NBAD into β-alanine and dopamine. Sclerotin also functions as a sclerotizing agent, i.e., it provides rigidness to insect cuticles [11, 12]. Previously generated *G. bimaculatus lac2* mutants with genome-editing displayed transparent cuticles and died within a few days after hatching [13], indicating that, similar to other insects, cuticle pigments in crickets are mostly melanin and sclerotin.

The importance of *ebony* and *tan* for cuticle pigmentation has been reported in other insects. On the one hand, knockout and/or knockdown of *ebony* leads to a black body color in the holometabolous insects *D. melanogaster*, *T. casteneum*, and *Henosepilachna vigintioctopunctata*, and in the hemimetabolous insects *Periplaneta americana* and *Oncopeltus fasciatus* [14–17]. In these insects, *ebony* systemically determines body color. On the other hand, *tan* is a suppression factor during NBAD synthesis. Mutations in *tan* reduced melanization in *D. melanogaster* [18]. In *O. fasciatus*, *tan* knockdown reduces the level of melanin pigmentation in a limited area [17]. The function of *tan* on the body color of insects likely varies among species.

In this study, using CRISPR/Cas9, we generated homozygous knockout mutants of *G. bimaculatus* (*Gb*) *ebony* and *tan*, which we used to analyze the *in vivo* functions of these genes. Additionally, we analyzed the changes in the expression levels of the *Gb'ebony* and *Gb'tan* transcripts in wild-type crickets and compared them with body color patterns. Our results provide

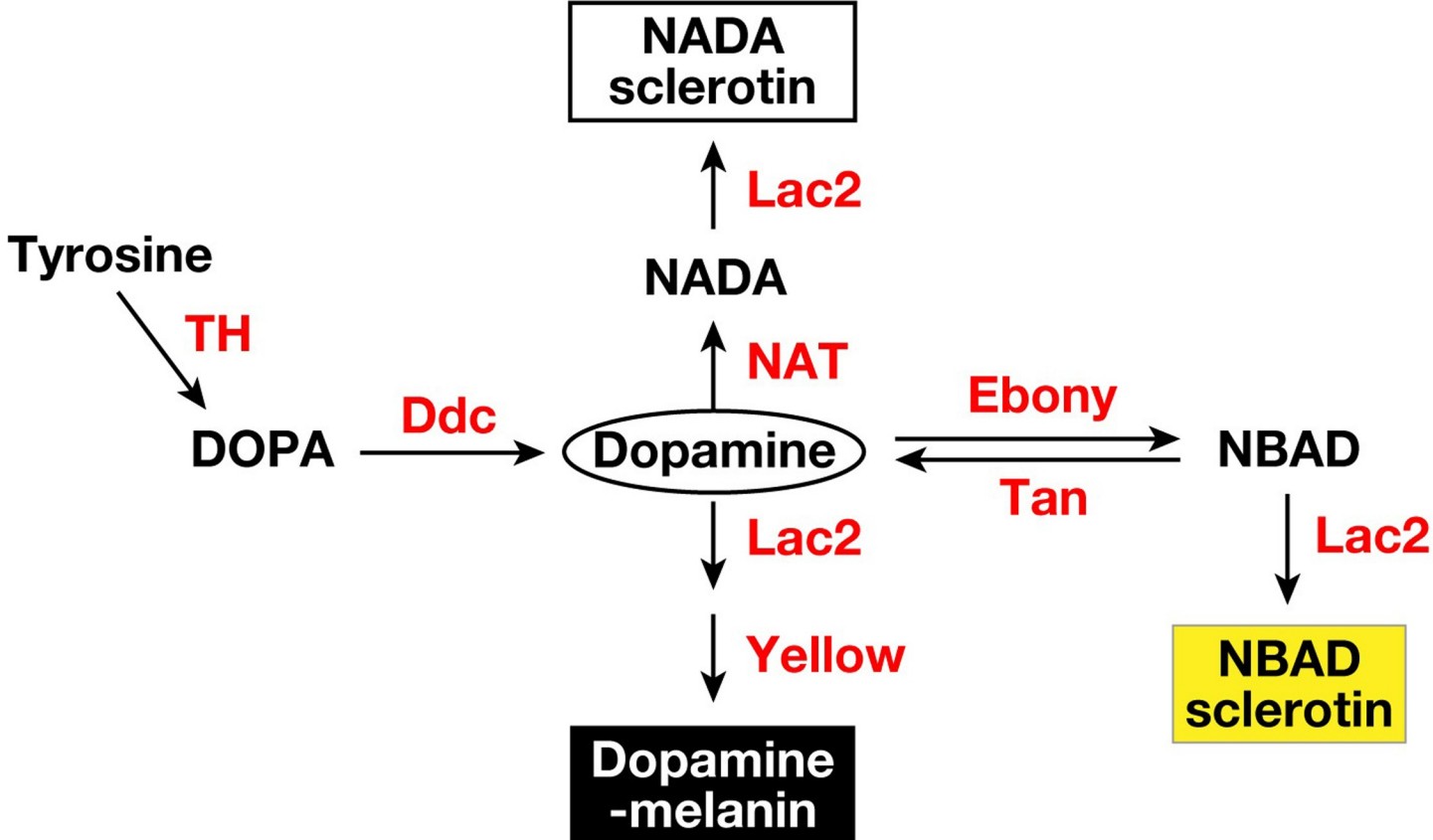

**Fig 1. Pathway of melanin and sclerotin biosynthesis in insects.** Dopamine, which is synthesized from tyrosine by the activity of TH and Ddc, becomes three pigments. Dopamine-melanin is produced by the activity of Yellow and Lac2. Excess dopamine is metabolized into NBAD and NADA by the activity of Ebony and NAT, respectively, to produce two types of sclerotins: yellow NBAD sclerotin and colorless NADA sclerotin. This pathway also includes Tan, an NBAD hydroxylase, which may coordinate the melanization level with Ebony. The metabolic pathway depicted was modified from Arakane et al. (2009) [10]. Intermediate products and pathways of DOPA-melanin biosynthesis have been omitted to demonstrate Dopamine-related pigmentation. Enzymes and metabolites are colored red and black, respectively. TH, Tyrosine hydroxylase; Ddc, DOPA decarboxylase; Ebony, NBAD synthase; Tan, NBAD hydroxylase; NAT, dopamine *N*-acetyltransferase; Yellow, Yellow protein.

evidence for the importance of *Gb'ebony* and *Gb'tan* in generating body color variation in the nymph and adult stages of the cricket.

## Results

### Identification of *Gryllus ebony* and *tan*

Using the sequences of *ebony* and *tan* from other insects as references, we searched for similar sequences in the *G. bimaculatus* genome (GenBank: GCA_017312745.1). The resulting information was used to clone 2,595 bp of *Gb'ebony* cDNA, including the full-length coding sequence. *Gb'ebony* was encoded on scaffold 146 of the *G. bimaculatus* genome throughout 15 exons (Fig 2A). The deduced *Gb'*Ebony protein consists of 864 amino acids with a molecular weight of 89,890. The primary structure of *Gb'*Ebony contains an AMP-binding domain located at 34–448 amino acids in the protein sequence. The results of phylogenic analysis of the amino acid sequences of Ebony proteins from *Gryllus* and other insects (aminoadipate-semialdehyde dehydrogenase (Aasdh) from *D. melanogaster*, a paralog of *Drosophila* ebony, was included as an outgroup) show that *Gb'ebony* is an ortholog of the *ebony* gene of other insects (Fig 2B).

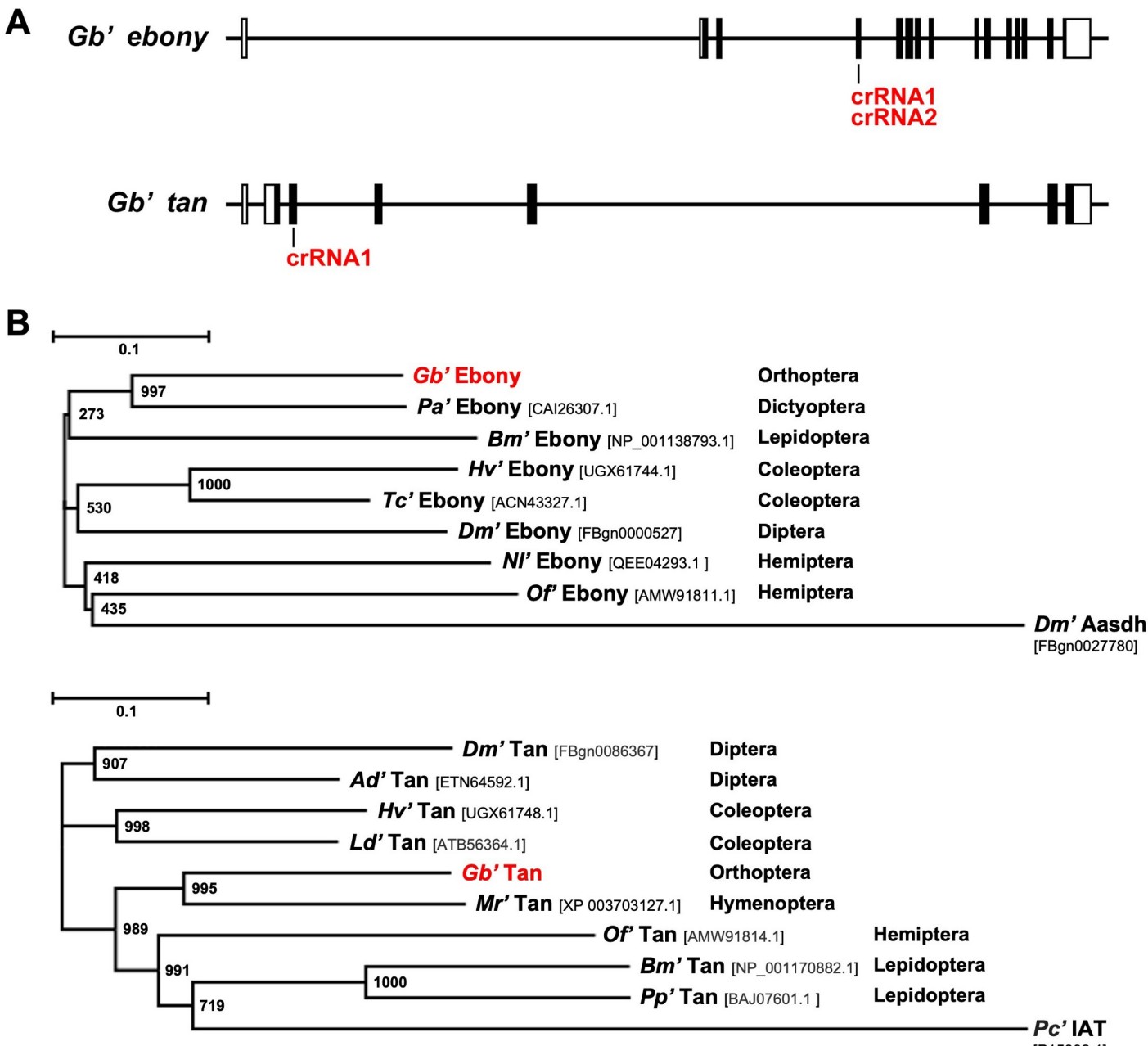

**Fig 2. Identification of *Gryllus ebony* and *tan*.** **(A)** Genomic structures of *Gb'ebony* and *Gb'tan*. Boxes indicate exons, and lines connecting boxes indicate introns. Black and white boxes indicate coding regions and untranslated regions, respectively. **(B)** The phylogenetic trees of the Ebony and Tan proteins from *Gryllus* and other insects. The sequences were aligned using the ClustalW program, and the phylogenetic tree was generated by a neighbor-joining method. *Pa: Periplaneta americana, Bm; Bombyx mori, Hv; Henosepilachna vigintioctopunctata, Tc; Tribolium castaneum, Dm; Drosophila melanogaster, Nl; Nilaparvata lugens, Of; Oncopeltus fasciatus, Ad; Anopheles darlingi, Ld; Leptinotarsa decemlineata, Mr; Megachile rotundata, Pp; Papilio polytes, Pc; Penicillium chrysogenum.*

We also cloned 1,455 bp of *Gb'tan* cDNA, which includes the full-length coding sequence. *Gb'tan* was encoded on scaffold 11 of the *G. bimaculatus* genome throughout 7 exons (Fig 2A). The deduced *Gb'*Tan protein consists of 393 amino acids with a molecular weight of 43,118. The primary feature of *Gb'*Tan is a 6-aminopenicillanic acid acyltransferase domain (Pfam03417) located at 127–376 amino acids in the deduced protein sequence. The results of

the phylogenic analysis of the amino acid sequences of Tan from *Gryllus* and other insects (iso-penicillin *N*-acyltransferase (IAT) from fungi *Penicillium chrysogenum*, the ancestral protein of insect Tan, was included as an outgroup) show that *Gb'tan* is an ortholog of the *tan* gene of other insects (Fig 2B). The sequences of *Gb'ebony* and *Gb'tan* were deposited into the DNA Data Bank of Japan under accession no. LC733201(*Gb'ebony*) and LC733202 (*Gb'tan*).

## Expression profiles of *Gb'ebony* and *Gb'tan* transcripts from the embryo through adult stages

To examine the relationship between the *Gb'ebony* and *Gb'tan* transcripts and cuticle pigmentation in the cricket, we analyzed the expression profiles of these genes from the embryo through the adult stages. *Gb'ebony* transcripts were detected 7 days after egg laying, and transcript levels increased immediately after the first instar nymphs hatched (Fig 3A). The expression level of the *Gb'ebony* transcript decreased rapidly after the completion of pigmentation a few hours after hatching and molting (compare the filled and unfilled points at D1 in the first to third instars enclosed with a red dotted frame in Fig 3A). The expression level before and after pigmentation in first instar nymphs showed a significant difference ($P < 0.01$). The expression level decreased after the peak expression in first instar nymphs and increased again in fifth instar nymphs. In the fifth instar and subsequent stages, a high level of expression immediately after molting and a subsequent decrease in expression were observed, and the peak expression tended to increase stage-to-stage. In the later stages, from the sixth to eighth instars, the expression levels of *Gb'ebony* transcripts varied greatly between individual crickets. At the seventh instar stage, expression in males tended to be higher than that in females. *Gb'ebony* expression was undetermined (Ct > 35) in several samples obtained from day 3 of the fifth, seventh, and eighth instars. Additionally, the expression level of *Gb'ebony* in adult males was significantly higher than that in adult females.

*Gb'tan* transcripts were detected 5 days after egg laying, and their levels increased immediately after hatching and each molting (Fig 3B). Gene expression decreased rapidly with the completion of melanin pigmentation in a few hours on day 1 of the first to third instar nymphs (compare the filled and unfilled points at D1 in the first to third instars enclosed with a red dotted frame in Fig 3B). There was a significant difference in gene expression, especially before and after the completion of melanin pigmentation on day 1 of the first ($P < 0.05$) and second ($P < 0.01$) instar nymphs. The highest expression level of *Gb'tan* occurred in first instar nymphs and gradually decreased to a peak of expression with each developmental stage. We observed no sex differences in the expression levels of *Gb'tan* transcripts.

## CRISPR/Cas9-based generation of *Gb'ebony* and *Gb'tan* homozygous knockout mutants

The *in vivo* functions of *Gb'ebony* and *Gb'tan* were investigated by generating homozygous knockout mutants using CRISPR/Cas9. We designed two guide RNAs (gRNAs) that bind to the third exon of *Gb'ebony* and one gRNA that binds to the second exon of *Gb'tan* (Fig 2A). Although only a single gRNA was used for *Gb'tan* knockout, its specificity was verified with the BLASTN program to search for matches within the *Gryllus* genome (GenBank: GCA_017312745.1) (S1 Fig). After injecting the Cas9-gRNA complex into the eggs, site-directed mutagenesis was detected using a mismatch-specific endonuclease, Guide-it Resolvase (Takara Bio, Shiga, Japan). As shown in Fig 4A, the DNA bands were digested when the PCR products from G0 eggs were incubated with the endonuclease. PCR products from the wild-type were not cleaved by the endonuclease. Raw data from this experiment can be found

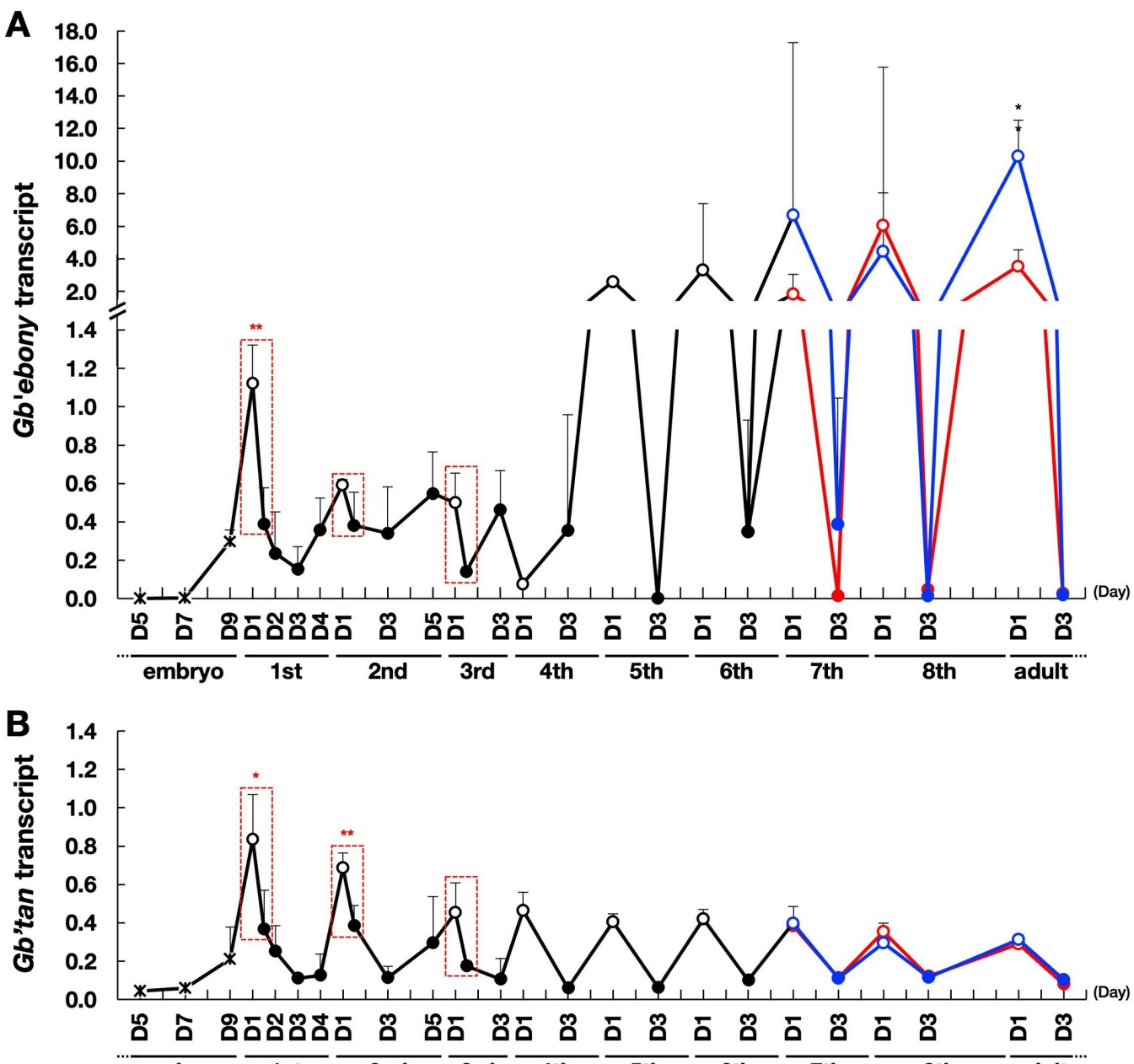

**Fig 3. Expression profiles of *Gb'ebony* and *Gb'tan* transcripts from the embryo through adult stages.** Relative expression levels of *Gb'ebony* (**A**) and *Gb'tan* (**B**) transcripts in whole embryos (D; day after egg laying) and the whole body of nymphs and adults (D; day after hatching or molting) were analyzed by RT-qPCR. *Gb'β-actin* was used as an internal control gene. Black, blue, and red lines indicate unsexed, male, and female crickets, respectively. Filled and unfilled points indicate pigmented and unpigmented crickets, respectively. The scale on the X-axis indicates one day. The first to third instars showed complete pigmentation within a few hours after hatching and molting. Gene expression in unpigmented and pigmented crickets on day 1 of each stage was analyzed using cDNA samples derived from crickets within 1 and 2–15 h, respectively, after hatching or molting (enclosed with a red dotted frame). Relative expression levels were calculated based on the amounts of transcripts in the first instar nymphs immediately after hatching. The data presented are the mean and SD (N ≥ 3). The lower sides of the error bars were omitted. The asterisks (*) and (**) mean $P < 0.05$ and $P < 0.01$, respectively, based on Student's *t*-test. Asterisks for the first to third instars are shown for significant differences in gene expression levels between unpigmented and pigmented crickets on day 1 of each stage. Asterisks from the seventh instar onward are indicated when there was a significant difference in gene expression between males and females.

in S1 Raw images. Furthermore, sequencing analysis confirmed the introduction of mutations in the gRNA target region of G0 eggs (S2 Fig).

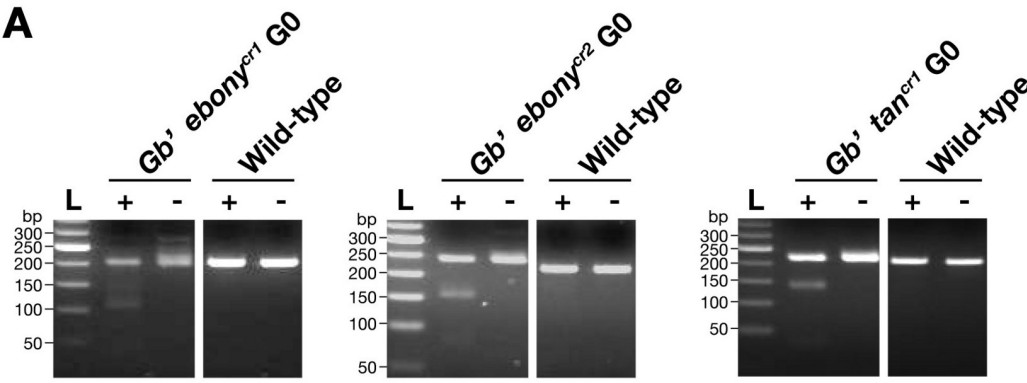

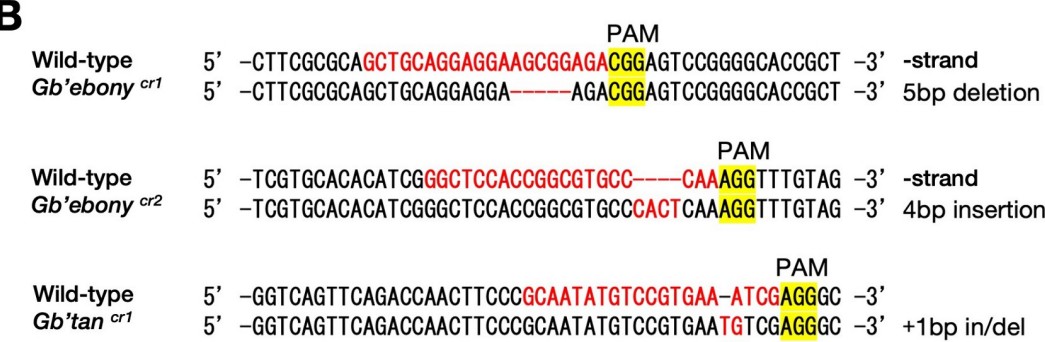

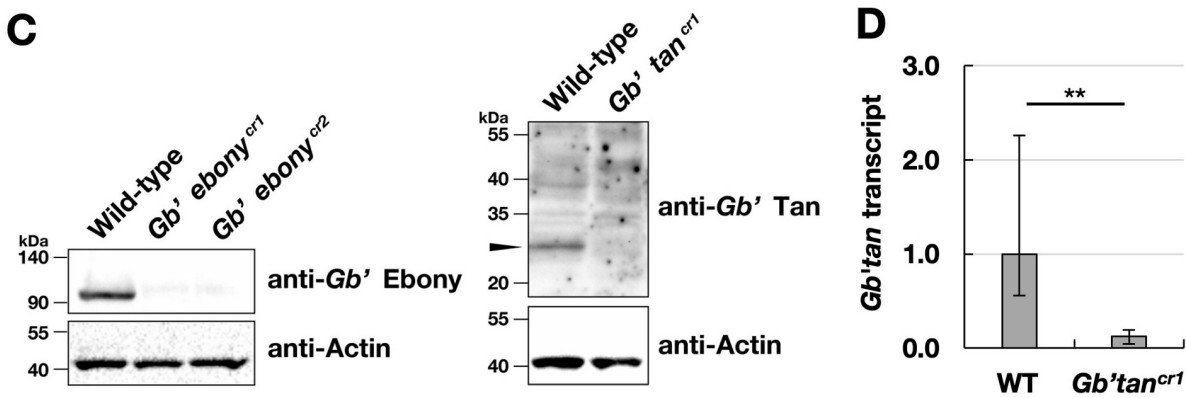

**Fig 4. The generation of *Gb'ebony* and *Gb'tan* homozygous mutants using the CRISPR/Cas9 system. (A)** Guide-it Resolvase assay for CRISPR/Cas9 mutagenesis. Gel images show the results of the assay performed on eggs after injection of the Cas9-gRNA complex. Minus lanes indicate the no Guide-it Resolvase control. Digested bands were detected when the reactions were performed with the enzyme (plus lanes). L: 50 bp DNA ladder. **(B)** Sequences of *Gb'ebony* and *Gb'tan* mutants generated by the CRISPR/Cas9 system. **(C)** Immunoblot analysis confirming the knockout of *Gb'ebony* and *Gb'tan* at the protein level. Antibodies against the recombinant His-tagged *Gb'*Ebony and *Gb'*Tan proteins were used in these experiments. Crude extracts from *G. bimaculatus* were separated by SDS-PAGE. *Gb'*Ebony and *Gb'*Tan proteins (arrowhead) were detected only in wild-type samples. β-Actin was included as a loading control. **(D)** Effects of *Gb'tan* knockout on *Gb'tan* transcription. RNA was extracted from the whole body of day 1 seventh instar nymphs of the wild-type and *Gb'tan* mutants within 1 h of molting and subjected to RT-qPCR. Data are presented as the mean value ± SD obtained from four biological replicates and three technical replicates. The asterisks (**) mean statistical significance at $P < 0.01$ in a Student's *t*-test.

We reared G0 insects and crossed them with wild-type crickets. The resulting eggs were genotyped, and the G1 heterozygous offspring of *Gb'ebony* and *Gb'tan* were sequenced. Male

and female mutants with the same type of frameshift mutation were isolated and crossed. Further sequence analysis of the gRNA-targeted region in the G2 offspring showed that the *Gb'ebony* frameshift mutants harbored a homozygous 5 bp deletion (*Gb'ebony* <sup>cr1</sup>) and a 4 bp insertion (*Gb'ebony* <sup>cr2</sup>) (Fig 4B). Meanwhile, the *Gb'tan* frameshift mutant carried a homozygous +1 bp insertion/deletion (*Gb'tan* <sup>cr1</sup>) (Fig 4B). Only one *Gb'tan* mutant strain was generated in this study, but mutations in off-target sites in *Gb'tan* crRNA1 (S1 Fig) were not observed in sequence analysis (S3 Fig).

To confirm whether the functions of *Gb'ebony* and *Gb'tan* in the mutants were completely disrupted at the protein level, we performed immunoblotting with polyclonal antibodies against the *Gb'*Ebony and *Gb'*Tan proteins. The anti-*Gb'*Ebony antibody recognized a 90-kDa protein in the wild-type, which corresponds to the molecular mass of the *Gb'*Ebony protein (95,963, calculated based on the estimated full-length coding sequence) (Fig 4C). However, the *Gb'*Ebony protein was not detected in the *Gb'ebony* <sup>cr1</sup> and *Gb'ebony* <sup>cr2</sup> mutants. Immunoblotting with anti-*Gb'*Tan detected a 25-kDa protein in the wild-type (Fig 4C). This protein band was not detected in the *Gb'tan* <sup>cr1</sup> mutant. However, the size of this band was smaller than the expected molecular mass of the *Gb'*Tan protein (43,118). The primary structure of the *Gb'*Tan protein shows homology with the acyl-coenzyme A:6-aminopenicillanic acid acyltransferase, a member of the peptidase C45 family. IAT from *P. chrysogenum*, which also belongs to the peptidase C45 family, is produced as an inactive precursor and matures through posttranslational self-cleavage at the Gry102-Cys103 peptide bond, resulting in 11 and 29 kDa α- and β-subunits, respectively [19]. In *D. melanogaster*, the 45-kDa Tan pre-protein is self-processed at Gry121-Cys122 into 30- and 15-kDa α- and β-subunits, respectively [20]. We confirmed that the self-cleavage site in *D. melanogaster* Tan was conserved in the *Gb'*Tan protein at Gry121-Cys122 (S4 Fig). Cleavage of the *Gb'*Tan protein at the same site should produce a 29.7-kDa protein with an active 6-aminopenicillanic acid acyltransferase domain (Pfam03417). This result suggests that the 25-kDa *Gb'*Tan protein is probably the mature form. To further validate the *Gb'tan* knockout, the transcript levels of *Gb'tan* were analyzed by RT-qPCR. The results showed that the amount of *Gb'tan* transcripts in the *Gb'tan* mutant line was significantly reduced to about one-tenth of that in the wild-type ($P < 0.01$, Fig 4D), demonstrating that mutagenesis was effective at the transcriptional level. Overall, these results indicate that the *Gb'ebony* and *Gb'tan* homozygous knockout mutants completely lost their functions.

## Phenotypes of homozygous *Gb'ebony* knockout mutants

We predicted that the *Gb'ebony* knockout mutation would cause a loss of yellow pigment because the gene encodes NBAD synthase, which is required for the synthesis of yellow sclerotin. Adult *Gb'ebony*<sup>cr1</sup> mutants had yellow-colored regions that were fully darkened compared to the wild-type, especially in the legs and forewings (Fig 5A). In the wild-type, the central region of the forewing was lightly transparent. This region in the *Gb'ebony* mutants was colored black and opaque. Fig 5B shows a region surrounded by two wing veins (arrowhead) and the upper edge of the hindwing that were colored black (Fig 5B). These black pigments may be dopamine-melanin.

*G. bimaculatus* is commonly known as the two-spotted cricket, a name derived from the spotted pattern on its forewing. In wild-type adults, the color of the anterior side of the forewing was lighter than that of the other side, producing a spotted pattern (Fig 5A). In *Gb'ebony* mutants, surprisingly, this pattern was colored white (probably due to colorless NADA sclerotin), although other areas of the forewing were dark. These results indicate that *Gb'ebony* is essential for the synthesis of yellow pigments, and its knockout causes systemic darkening of

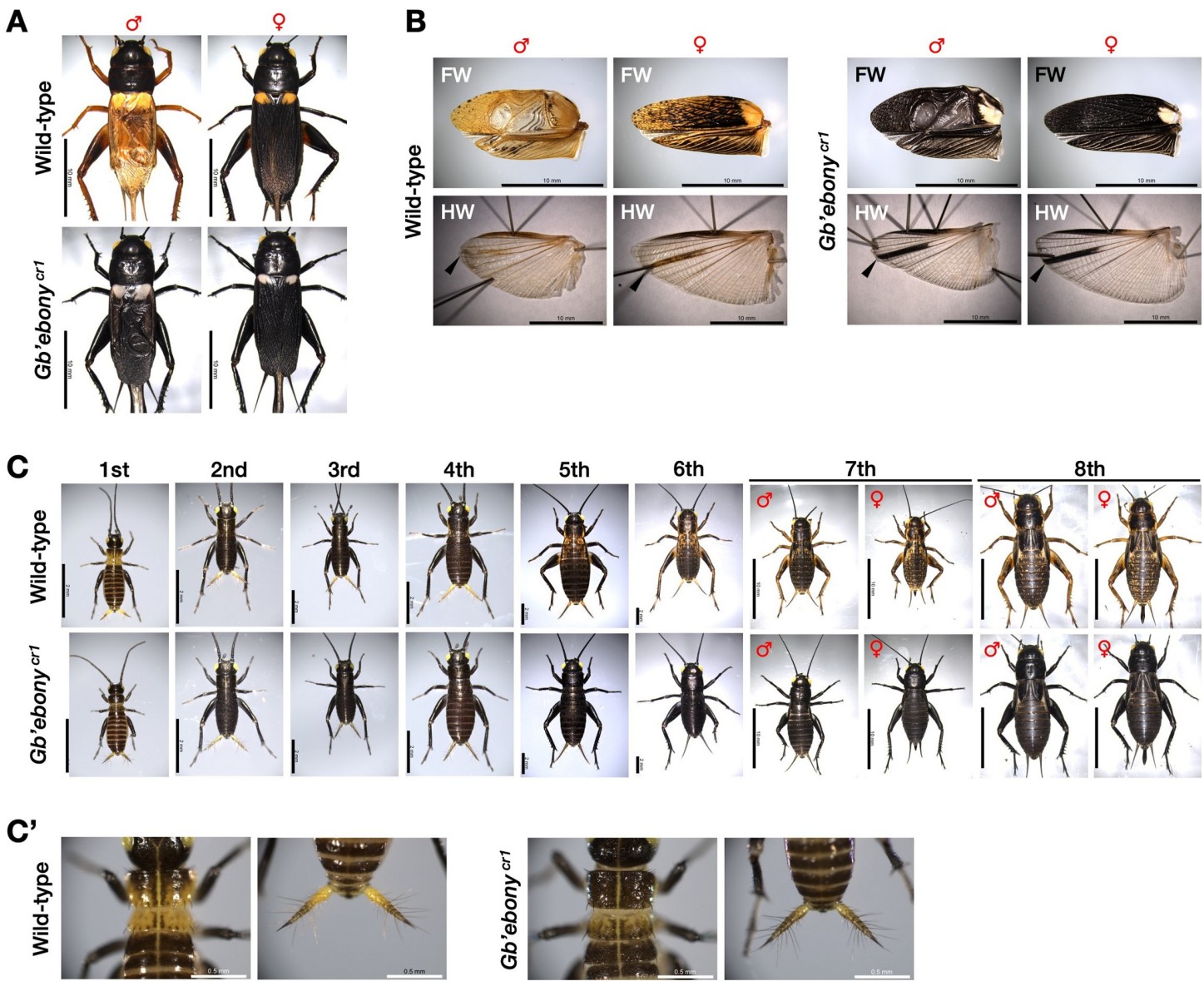

**Fig 5. Phenotype of *Gb'ebony* homozygous mutants. (A)** Dorsal views of wild-type and *Gb'ebony* $^{cr1}$ mutant adults. **(B)** Effect of *Gb'ebony* knockout on the color of adult wings. FW: Forewing, HW: Hind wing. **(C)** Dorsal views of wild-type and *Gb'ebony* $^{cr1}$ mutant nymph stages. **(C')** Magnified image of the dorsal side of the thorax and the tail in first instar nymphs. Scale bars: 10 mm in A and B; 2 mm (1st–6th instar nymphs) and 10 mm (7th–8th instar nymphs) in C; 0.5 mm in C'.

body color. In the nymphal stages, the dorsal side of the thorax segments 1 and 2 and the tail surface in the first instar nymphs were yellow ([Fig 5C and 5C']). In *Gb'ebony* mutants, these regions were melanized. In the second to fourth instar nymphs, the change in the body color of the *Gb'ebony* mutants was limited because the wild-type originally has a dark black body color. Starting from the sixth instar stage, the body color of the wild-type became increasingly yellow until the eighth (ultimate) stage. In contrast, the yellow pigment of the *Gb'ebony* mutants was lost during these stages. These results indicate that *Gb'ebony* is required for the systemic determination of body color. We also analyzed the *Gb'ebony* $^{cr2}$ strain, which was generated using a different gRNA (crRNA2) than that used in the generation of the *Gb'ebony* $^{cr1}$ strain. The *Gb'ebony* $^{cr2}$ mutants exhibited systemic darkening of body color at all stages ([S5]

Fig), indicating that there were no phenotypic differences between the two mutant strains. This result also indicates that there were no off-target effects on the phenotype.

### Phenotypes of homozygous *Gb'tan* knockout mutants

*Gb'tan* encodes NBAD hydroxylase, which degrades NBAD into dopamine and β-alanine, a function opposite to that of NBAD synthase, which is encoded by *Gb'ebony*. Therefore, we predicted that the *Gb'tan* knockout would enhance NBAD yellow sclerotin synthesis and reduce melanin pigmentation. However, as seen in Fig 6A, the change in the body color of adult *Gb'tan* $^{cr1}$ mutants was comparable to that of the wild-type. Still, in the isolated male forewing of the adult *Gb'tan* $^{cr1}$ mutants, the melanin pigmentation level was slightly reduced, and light transparency was enhanced (Fig 6B), indicating that the synthesis of dopamine-melanin was inhibited by the *Gb'tan* knockout. The body color of *Gb'tan* mutants at the seventh instar appeared remarkably lighter than that of the wild-type (Fig 6C). To quantify the color difference between the wild-type and *Gb'tan* mutants, we measured the mean grayscale intensity, consisting of 256 tones of color gradients, with 0 and 255 indicating white and black, respectively, at the penultimate nymph stage. A lower and higher intensity value indicates a darker and lighter body color, respectively. In males, the mean intensities of the wild-type and *Gb'tan* mutants were 77 and 96, respectively, significantly different ($P < 0.01$) (Fig 6D). In females, the mean intensities of the wild-type and *Gb'tan* mutants were 84 and 96, respectively (Fig 6D). This also showed a significant difference ($P < 0.05$), but less different than in males. These results indicate that *Gb'tan* is essential for proper melanin pigmentation in limited developmental stages and tissues, and its knockout enhances the synthesis of NBAD yellow sclerotin.

## Discussion

Insect dopamine is an important substrate for the biosynthesis of pigments such as dopamine-melanin, NBAD sclerotin, and NADA sclerotin. In this study, we focused on the function of *G. bimaculatus ebony* and *tan*, which are involved in dopamine metabolism.

### The expression of the *Gb'ebony* and *Gb'tan* transcripts during hatching and molting

We first examined the expression profiles of *Gb'ebony* and *Gb'tan* during development to assess their relationship with cuticle pigmentation. After hatching and molting, crickets (re) construct body color on the cuticle within a few hours. During these periods, the expression levels of *Gb'ebony* and *Gb'tan* were elevated and then decreased after pigmentation was completed (Fig 3). These results indicate that *Gb'ebony* and *Gb'tan* are involved in cuticle pigmentation in crickets. Similarly, in *H. vigintioctopunctata (Hv)*, the expression of *Hvtan* and *Hvebony* is upregulated before and/or immediately after molting [16].

Insect metamorphosis is rigidly regulated by two molting hormones: juvenile hormone (JH) and 20-hydroxyecdysone (20E) [21]. High levels of both 20E and JH occur when nymphs are molting, whereas only the 20E titer is high during metamorphosis into the adult stage. In *G. bimaculatus*, the levels of JH and *Gb'E93* (an ecdysone-induced protein) increase periodically during molting [22, 23], following similar patterns as that of *Gb'ebony* and *Gb'tan*. Additionally, in *Bombyx mori*, *BmDdc* is expressed in response to exogenously administered phytogenous ecdysteroids [24]. Overall, these results indicate that pigment metabolism genes, including *Gb'ebony* and *Gb'tan*, are regulated by the molting hormone signal cascade.

Molting is accomplished in three phases: the preparatory, cuticle induction, and ecdysial phases [21]. These phases are regulated by 20E; for example, the initiation of the cuticle

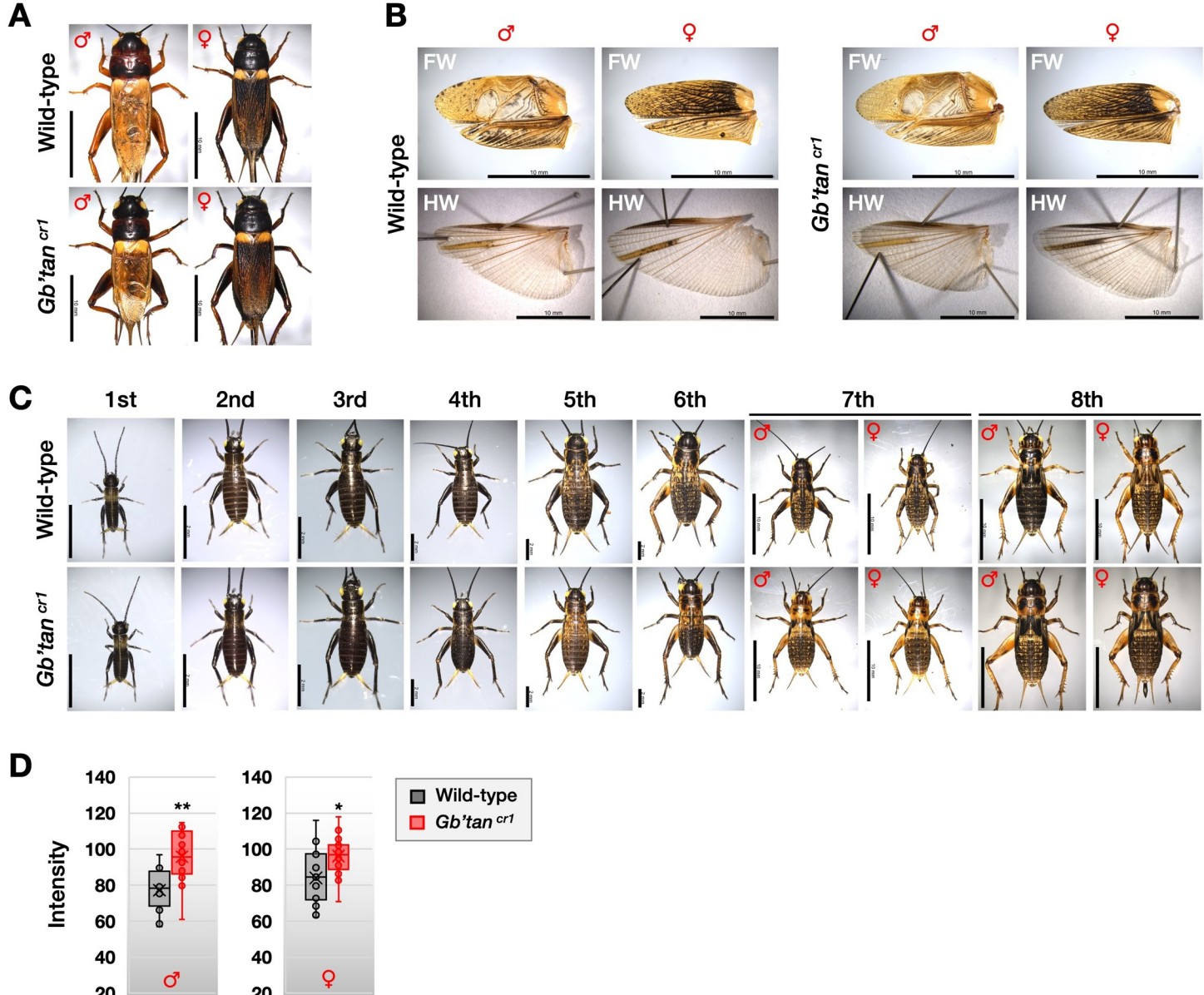

**Fig 6. Phenotype of *Gb'tan* homozygous mutants.** **(A)** Dorsal views of wild-type and *Gb'tan* $^{cr1}$ mutant adults. **(B)** Effect of *Gb'tan* knockout on the color of adult wings. FW: Forewing, HW: Hindwing. **(C)** Dorsal views of wild-type and *Gb'tan* $^{cr1}$ mutant nymph stages. **(D)** Quantification of grayscale intensity in wild-type and *Gb'tan* mutants at the seventh instar stage. The intensity of body color excluding the appendages was measured using ImageJ Fiji software (https://fiji.sc). The grayscale intensity consists of 256 tones of color gradients, with 0 and 255 indicating white and black, respectively. The asterisks (*) and (**) mean $P < 0.01$ and $P < 0.001$, respectively, based on a Mann–Whitney U test (N ≧ 12). Scale bars: 10 mm in A and B; 2 mm (1st–6th instar nymphs) and 10 mm (7th–8th instar nymphs) in C.

induction phase requires a high 20E titer. During the cuticle induction phase, *Drosophila* express the ecdysone-induced transcription factors *E74A* and *E75B*. In *Tribolium*, *E75* is required for adult metamorphosis, and its RNAi knockdown causes an enhanced expression of *Th*, the gene encoding tyrosine hydroxylase, which is the initial enzyme in dopamine synthesis [25]. The expression of *E75* is suppressed by JH-mimic treatment. These findings from other studies suggest that cuticle pigmentation is controlled by 20E and JH upon molting. However, the mechanism that regulates *Gb'ebony* and *Gb'tan* during molting remains unknown.

## Role of *Gb'ebony* in cuticle pigmentation

*Gb'ebony* encodes NBAD synthase, which catalyzes the conjugation of dopamine with β-alanine to produce NBAD, a precursor of the yellow pigment. We generated two loss-of-function mutants for *Gb'ebony* (*Gb'ebony^{cr1}* and *Gb'ebony^{cr2}*) using CRISPR/Cas9 genome editing and revealed the role of *Gb'ebony* in the cuticle pigmentation of crickets. The complete loss-of-function of *Gb'ebony* in the mutants was verified at the protein level (Fig 4C). Mutations in *Gb'ebony* resulted in the darkening of the yellow-colored body regions, except for the spotted patterns on the forewing (Fig 5). The darkening of the body color in *Gb'ebony* may be due to an over-accumulation of dopamine derived from the excess of intracellular dopamine resulting from disrupted NBAD synthesis. Our data clearly demonstrate that the yellow pigment on the cuticle of *G. bimaculatus* is NBAD yellow sclerotin produced by *Gb'ebony*, and knockout of *Gb'ebony* results in the systemic darkening of body color. Similarly, the role of *ebony* in the cuticle pigmentation of *Drosophila* results in the systemic darkening of body color, as demonstrated in a genetic *ebony* knockout strain [14]. In adult *Oncopeltus* and *Periplaneta*, depletion of *ebony* results in the systemic darkening of body color, although in *Oncopeltus*, the color of the hindwing remains unchanged [17]. Similar body regions are affected by *Gb'ebony* mutations in *Periplaneta* and *Drosophila*. Overall, these results indicate that the systemic function of *ebony* in cuticle pigmentation is conserved in a broad range of insect species.

In this study, the *Gb'ebony* mutants displayed a white spotted pattern in the forewing. The white pigment is probably NADA sclerotin. In the absence of *Gb'ebony*, the NADA and melanin pathways compete because both use dopamine as their substrate. Therefore, there are two possible mechanisms for the formation of a white spotted pattern in the forewing. First, the expression of the melanin synthesis gene *yellow* is lost in the spots; and second (which we believe is more likely), the expression of the NADA synthesis gene *nat* is enhanced. The results of recent studies on other insects provide support for the second possibility. For example, *aaNAT* is highly expressed in the white spots of the hemelytra of *Platymeris*. RNAi knockdown of this gene results in the blackening of these spots [26]. These results suggest that the expression of the pigment metabolism gene in the forewing spots of *Gb'ebony* mutants is regulated differently than other body parts, and *aaNAT* is probably required for the white-spotted patterning in crickets. However, the mechanism behind the spatial expression of pigment metabolism genes remains unknown.

## Role of *Gb'tan* in cuticle pigmentation

*Gb'tan* encodes NBAD hydroxylase, which catalyzes the degradation of NBAD into dopamine and β-alanine, an activity that is opposite to that of the *Gb'ebony* gene product. We predicted that *Gb'tan* knockout would enhance NBAD yellow sclerotin synthesis. We then generated loss-of-function mutants for *Gb'tan* (*Gb'tan^{cr1}*) to test our prediction. In such mutants, the body color of late-stage nymphs turned a bright yellow color (Fig 6), which is probably due to an over-production of NBAD yellow sclerotin. Our data indicate that *Gb'tan* is required for proper melanization and that its knockout results in enhanced NBAD yellow sclerotin synthesis. The level of the contribution of the *tan* gene to cuticle pigmentation in different insect species varies. In *Drosophila*, the reduction in melanization of *tan* mutants spans the entire body [18, 27]. In contrast, *Oncopeltus* adults with depleted *tan* via RNAi display no significant alteration in body color, although the depletion of *ebony* results in the darkening of body color [17]. Liu et al. (2016) [17] proposed that in hemimetabolous insects, either *ebony* or *tan* is required for cuticle pigmentation. Our results mostly agree with this, as the body color of *Gb'tan* mutants clearly changed only at the seventh nymph stage. However, the reason body color changed during these stages remains unknown. The seventh instar stage shows a

relatively low *Gb'tan* expression (Fig 3B), implying that factors other than *Gb'tan* expression level may also be involved in the increased appearance of the knockout phenotype.

One possibility is that differences in NBAD levels *in vivo* at various developmental stages may be involved in the knockout phenotype of *Gb'tan*, since *Gb'*Tan uses NBAD as a substrate. At the seventh instar stage, *Gb'ebony* expression reaches approximately peak levels (Fig 3A), implying that NBAD levels may have peaked as well. This means that the contribution of *Gb'tan* is particularly high at this stage, and the knockout phenotype has emerged. This hypothesis is also supported by data showing that wild-type males express more *Gb'ebony* in seventh instar nymphs and adults (Fig 3A) and that males of *Gb'tan* mutants at this stage have a more clearly observable phenotype than females (Fig 6). The possibility that *ebony* function contributes to differences in NBAD levels *in vivo* during development has been suggested in other insect species; Ze et al. (2022) [16] reported that NBAD may be more abundant in adults of *H. vigintioctopunctata*.

## The mechanism determining body color in *Gryllus*

In this study, we investigated the function of *Gb'ebony* and *Gb'tan* in the metabolism of dopamine and NBAD by generating homozygous knockout mutants through CRISPR/Cas9 genome editing. The body color of *Gb'ebony* mutants showed a systemic darkening, which was probably the result of dopamine-melanin produced from excess dopamine, indicating that the loss-of-function of *Gb'ebony* results in enhanced dopamine-melanin synthesis. Meanwhile, the body color of the *Gb'tan* mutants was yellow, but the function of *Gb'tan* was not systemic. The yellow pigment is probably NBAD sclerotin, indicating that the loss-of-function of *Gb'tan* enhances the synthesis of NBAD sclerotin. As shown in Fig 7A, the results of our study indicate that the body color of crickets is generated by three cuticle pigments (dopamine-melanin, NBAD sclerotin, and NADA sclerotin), which are synthesized from dopamine. The dysfunction of one gene in this metabolic pathway caused a body color change due to an over-production of the other pigments.

A summary of the expression profiles of *Gb'ebony* and *Gb'tan* from eggs through adults is shown in Fig 7B. In first instar nymphs, *Gb'ebony* is upregulated after hatching, followed by low expression levels in the second to fourth instar nymphs. Then, the expression levels change periodically in the fifth instar stage through the adults, showing peak levels immediately after molting in each stage. *Gb'tan* is also expressed according to hatching, and its peak expression levels gradually decrease toward the adult stage. Cricket body colors also change through the stages: in the first instar nymph and the fifth instar to adult, the body color is relatively light, whereas in the second to fourth instar, the body color is relatively dark (Fig 7B). The dark pigmentation is thought to be caused by the accumulation of dopamine-melanin and the depletion of NBDA due to the low and high expression of *Gb'ebony* and *Gb'tan*, respectively. However, the light pigmentation probably results from the depletion of dopamine-melanin and the accumulation of NBAD due to high and low expression of *Gb'ebony* and *Gb'tan*, respectively. Our results also show that adult male crickets tend to be more yellow than females (Fig 5A). The *Gb'ebony* expression level was significantly higher in adult males than in females (Fig 3A), suggesting that the yellow color in male crickets is probably NBAD sclerotin that is generated by the high expression of *Gb'ebony*.

This study provides evidence that the combination of *Gb'ebony* and *Gb'tan* expression levels governs the generation of stage-specific body color patterns during postembryonic development. This finding provides new insights into the molecular mechanisms by which insects diversify their body coloration between and within species and adapt to their surroundings.

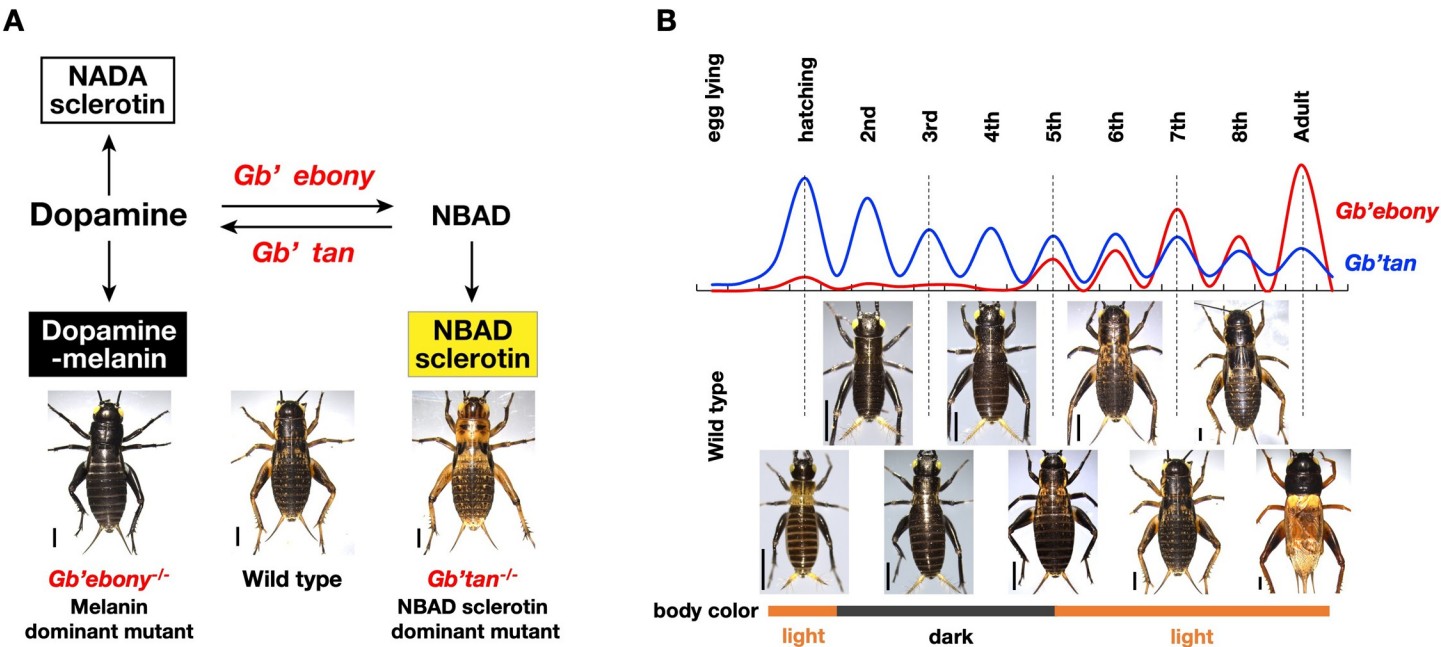

**Fig 7. Mechanism for determining body color in *G. bimaculatus*.** (A) Pathway of melanin and sclerotin biosynthesis in *G. bimaculatus*. Knockout of *Gb'ebony* (encoding NBAD synthase) and *Gb'tan* (encoding NBAD hydroxylase) causes body color changes due to over-production of melanin and NBAD sclerotin, respectively. (B) The correlation between the expression patterns of *Gb'ebony* and *Gb'tan* and the body color transition of wild-type crickets from the nymphs through adult stages. Scales of the Y-axis in the graph differ between *Gb'ebony* and *Gb'tan* for the comparison of periodic patterns. The images of the crickets in this figure were reprinted from Figs 5 and 6. Scale bars: 2 mm in A; 1 mm (1st–4th instar nymphs) and 2 mm (5th–8th instar nymphs and adult) in B.

## Materials and methods

### Animals

Nymphs and adults of the *G. bimaculatus* white-eyed mutant strain [28] were reared in plastic cases at 30˚C ± 1˚C and 30%–40% relative humidity under a 10 h light and 14 h dark photoperiod. They were fed artificial fish food (Kyorin, Hyogo, Japan).

### cDNA cloning

Total RNA was prepared using TRIzol Reagent (Thermo Fisher Scientific, Massachusetts, USA) and treated with ezDNase Enzyme (Thermo Fisher Scientific) according to the manufacturer's instructions. RNA was then reverse transcribed using a SuperScript IV First-Strand Synthesis System (Thermo Fisher Scientific) with oligo dT primers. *Gb'ebony* and *Gb'tan* cDNA, including the full-length coding sequences, were amplified from cDNA derived from third instar nymphs and adult forewings, respectively, using the primers listed in S1 Table. The PCR products were inserted into the pGEM T-Easy vector (Promega, Wisconsin, USA) and sequenced.

### CRISPR/Cas9-mediated genome editing

**Preparation of gRNAs and injection solution.** The Alt-R CRISPR Cas9 System (IDT, Iowa, USA) was used according to the manufacturer's instructions. CRISPR RNAs (crRNAs), which are specific for target genome sequences, were designed using the online tool CRISPRdirect (https://crispr.dbcls.jp) following the NGG protospacer-adjacent motif (PAM) [29]. The target sequences of the designed crRNAs are listed in S2 Table. The sequence specificity of the

crRNAs was checked using the BLASTN program against *G. bimaculatus* genomic data (Gen-Bank: GCA_017312745.1). The sequences of the designed crRNAs are listed in S2 Table. Each crRNA and trans-activating crRNAs was hybridized in duplex buffer (IDT) at a concentration of 29.7 μM. The 29.7 μM gRNA complex was then mixed with an equal volume of 12.4 μM Cas9 (Alt-R® S.p. HiFi Cas9 Nuclease V3; IDT) in the injection solution (1.4 mM NaCl, 0.07 mM Na$_2$HPO$_4$, 0.03 mM KH$_2$PO$_4$, and 4 mM KCl) and incubated for 15 min at room temperature. The mixture containing the Cas9-gRNA complex was then diluted 20-fold with injection solution and used for microinjection.

**Microinjection into cricket eggs.** Cricket eggs were microinjected according to previously reported methods [4, 30]. Briefly, the Cas9-gRNA complex was injected into cricket eggs (100–200 eggs) using a microinjector (IM-300 Microinjector; Narishige, Tokyo, Japan) and a compressor (0.2LE-8SBZN, Narishige). Injections were performed within 3 h after starting egg incubation. After the injections, the eggs were soaked in PBS containing penicillin and streptomycin and kept at 30°C for 2 d. Thereafter, the eggs were transferred on a paper towel that was constantly wetted with water and kept there until they hatched.

**Genotyping.** Genomic DNA was extracted from eggs or legs using the CicaGeneus Total DNA PrepKit (Kanto-chemical, Tokyo, Japan). Mutations at the gRNA-targeted-regions of *Gb'ebony* and *Gb'tan* in the G0 and G1 embryos were detected by amplifying these regions and digesting the PCR products with the Guide-it Mutation Detection Kit (Takara Bio, Japan) based on a previously reported method [13, 30]. Briefly, the PCR products were reannealed under the following conditions: 95°C for 5 min, 95°C–85°C at −2°C/sec, and 85°C–25°C at −0.1°C/sec. Next, 10 μL of the PCR products were mixed with 5 μL of distilled water and 1 μL of Guide-it Resolvase (Takara Bio) in the Guide-it Mutation Detection Kit (Takara Bio). This was incubated at 37°C for 15 min. Then, mutagenesis in the gRNA-targeted region was detected by electrophoresis using 3% agarose gel. Alternatively, sequence analysis of the gRNA-targeted-regions in the individual G1 and G2 nymphs was performed. The primers used are listed in S1 Table.

## RT-qPCR

All crickets were sampled during the dark period. Unpigmented and pigmented crickets on day 1 of each stage were sampled within 1 h and between 2–15 h, respectively, after hatching and molting. mRNA extraction and reverse transcription were performed from embryos, nymphs, and adult whole bodies according to the method described above (the cDNA cloning section). RT-qPCR was performed in 20 μL reaction mixtures, each containing 10 ng of template cDNA, 0.3 μM of the primers listed in S1 Table, and TB green Premix Ex TaqII (Tli RNaseH Plus; Takara Bio). RT-qPCR was performed using the QuantStudio3 Realtime PCR System (Thermo Fisher Scientific) with *Gb'β-actin* (Ishimaru et al., 2016) as an internal control. The amounts of *Gb'ebony* and *Gb'tan* mRNA were calculated according to the ΔΔCt method.

## Image acquisition and intensity measurement

Digital images of nymphs, adults, and isolated adult wings were taken using a DFC7000 T digital camera (Leica, Wetzlar, Germany) connected to an M165 FC stereomicroscope (Leica). Signal intensity, measured as the mean grayscale value on 8-bit images, was determined using the image processing software ImageJ Fiji (https://fiji.sc).

## Production of *Gb'*Ebony and *Gb'*Tan antibodies

To construct expression vectors of the His-tagged *Gb'*Ebony and *Gb'*Tan proteins in *Escherichia coli*, the coding sequences of *Gb'ebony* and *Gb'tan* were amplified using the primers listed in S1 Table and inserted into the *Nde* I site of the pET41-b vector using In-Fusion Snap

Assembly Master Mix (Takara Bio). *E. coli* strain BL21(DE3) was transformed with the expression vector and was then grown in Luria-Bertani medium containing kanamycin until the optical density at 600 nm reached 0.6. At this point, the expression of the recombinant protein was induced by adding 0.1 mM isopropyl β-D-1-thiogalactopyranoside to the culture and incubating it for 4 h at 37˚C. The cells were collected by centrifugation and resuspended in BugBuster (Merck Millipore, Massachusetts, USA) containing 5 mM EDTA and 1 mM phenyl-methylsulfonyl fluoride (PMSF)). The suspension was incubated for 15 min on ice, and the insoluble proteins were collected by centrifugation and dissolved in SDS sample buffer (62.5 mM Tris-HCl (pH6.8), 2% (w/v) SDS, 10% (v/v) glycerol, 0.001% (w/v) bromophenol blue). A rabbit and mouse were then immunized with the purified recombinant *Gb'*Ebony and *Gb'*Tan protein, respectively.

## Immunoblotting

The crude proteins, including *Gb'*Ebony and *Gb'*Tan, were prepared as follows: heads from 5–10 individual 2nd instar nymphs (for *Gb'*Ebony) and forewings from three individual adult male crickets (for *Gb'*Tan) were homogenized in 250 μL of ice-cold PBS (containing 1 mM PMSF and 5 mM EDTA) using Bio-masher III (Nippi, Tokyo, Japan). The homogenate was filtered by centrifugation, then 1 mL of acetone was added to the filtrate and incubated for 1 h at −20˚C. The samples were centrifuged, and the precipitated protein was dissolved in SDS sample buffer. The concentration of proteins was measured using the BCA Protein Assay Kit (Pierce, Massachusetts, USA).

β-mercaptoethanol was added to all the samples to a final concentration of 5% (v/v), and the mixture was boiled for 5 min. Equal amounts of protein (25–50 μg) were loaded into each lane of an SDS-PAGE gel with a 10%–20% gradient. Proteins in the gel were transferred onto a PVDF membrane and reacted with a rabbit anti-*Gb'*Ebony antisera (1:5,000), mouse anti-*Gb'*Tan antisera (1:5,000), and mouse β-Actin antibody (1:10,000) (66009-1-lg; Proteintech, Illinois, USA). Horseradish peroxidase (HRP)-conjugated anti-rabbit IgG antibody (1:10,000) (CST, Massachusetts, USA) or HRP-conjugated anti-mouse IgG antibody (1:10,000) (Proteintech) were used as the secondary antibodies. Signals were detected using ECL Prime (Cytiva, Tokyo, Japan) and visualized using the ChemiDoc XRS Plus (Bio-Rad, California, USA).

## Supporting information

**S1 Fig. The specificity of *Gb'tan* crRNA1.** The BLASTN program was used to search for candidates of off-target sequence effects of *Gb'tan* crRNA1 in the *Gryllus* genome (GenBank: GCA_017312745.1). In addition to *Gb'tan*, three other sequences were detected on Scaffold307 (GenBank: BOPP01000307.1), Scaffold70 (GenBank: BOPP01000070.1), and Scaffold18 (GenBank: BOPP01000018.1), but their sequences have no PAM (NGG) sequence.
(JPG)

**S2 Fig. PCR sequencing of G0 eggs.** Sequence analysis of mutations introduced around the crRNA target region (highlighted in yellow) in the *Gb'ebony*[cr1] (A), *Gb'ebony*[cr2] (B), and *Gb'tan*[cr1] (C) mutants of the G0 generation.
(JPG)

**S3 Fig. Sequence analysis of the potential off-target sites of crRNAs used to generate *Gb'tan* mutant strains.** PCR amplification and sequence analysis of off-target sites (highlighted in blue) predicted for *Gb'tan* crRNA1 ((**A**) Scaffold307 (GenBank: BOPP01000307.1), (**B**) Scaffold70 (GenBank: BOPP01000070.1), and (**C**) Scaffold18 (GenBank: BOPP01000018.1) on *G. bimaculatus* genomic data (GenBank: GCA_017312745.1))

were performed using the genome of the *Gb'tan* mutant as a template and the primers listed in S1 Table compared with the wild-type.
(JPG)

**S4 Fig. Predicted self-cleaving site in *Gb'*Tan protein.** Primary structures of *Gb'*Tan and *D. melanogaseter* Tan were aligned using the ClustalW program. The amino acid sequence of the self-cleaving site in *D. melanogaseter* Tan is conserved in the *Gb'*Tan protein.
(JPG)

**S5 Fig. Phenotype of the *Gb'ebony^{cr2}* mutant strain. (A)** Dorsal views of wild-type and *Gb'ebony^{cr2}* mutant adults. **(B)** Effect of *Gb'ebony* knockout on the color of adult wings. FW: Forewing, HW: Hind wing. **(C)** Dorsal views of wild-type and *Gb'ebony^{cr2}* mutant nymph stages. **(C')** Magnified image of the dorsal side of the thorax and the tail in first instar nymphs. See Fig 5 for a picture of the wild-type. Scale bars: 10 mm in A and B; 2 mm (1st–6th instar nymphs) and 10 mm (7th–8th instar nymphs) in C; 0.5 mm in C'.
(JPG)

**S1 Table. Primers used in this study.**
(DOCX)

**S2 Table. Target sequences of crRNA.** Underlined letters indicate the PAM sequence.
(DOCX)

**S1 Raw images.**
(JPG)

# Acknowledgments

We thank Kayoko Tada for her help in maintaining the cricket strains produced in this study.

# Author Contributions

**Conceptualization:** Takahito Watanabe, Taro Mito.

**Data curation:** Shintaro Inoue.

**Formal analysis:** Shintaro Inoue.

**Funding acquisition:** Takahito Watanabe, Katsuyuki Miyawaki, Takeshi Nikawa, Akira Takahashi, Taro Mito.

**Investigation:** Shintaro Inoue, Taiki Hamaguchi, Yoshiyasu Ishimaru.

**Methodology:** Takahito Watanabe, Taro Mito.

**Project administration:** Takahito Watanabe, Taro Mito.

**Supervision:** Taro Mito.

**Validation:** Katsuyuki Miyawaki, Takeshi Nikawa, Akira Takahashi, Sumihare Noji.

**Visualization:** Shintaro Inoue.

**Writing – original draft:** Shintaro Inoue, Taro Mito.

**Writing – review & editing:** Shintaro Inoue, Takahito Watanabe, Yoshiyasu Ishimaru, Katsuyuki Miyawaki, Takeshi Nikawa, Akira Takahashi, Sumihare Noji, Taro Mito.

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
