## [Decision Letter · Decision Letter 0]

16 Mar 2023

PONE-D-23-01091Combinatorial expression of *ebony* and *tan* generates body color variation from nymph through adult stages in the cricket, *Gryllus bimaculatus*PLOS ONE

Dear Dr. Mito,

Thank you for submitting your manuscript to PLOS ONE. After careful consideration, we feel that it has merit but does not fully meet PLOS ONE’s publication criteria as it currently stands. Therefore, we invite you to submit a revised version of the manuscript that addresses the points raised during the review process.

We look forward to receiving your revised manuscript.

Kind regards,

Peng He, Ph.D

Academic Editor

PLOS ONE

Journal Requirements:

Reviewers' comments:

Reviewer's Responses to Questions

**Comments to the Author**

1. Is the manuscript technically sound, and do the data support the conclusions?

Reviewer #1: Yes

Reviewer #2: Yes

2. Has the statistical analysis been performed appropriately and rigorously? 

Reviewer #1: Yes

Reviewer #2: Yes

3. Have the authors made all data underlying the findings in their manuscript fully available?

Reviewer #1: Yes

Reviewer #2: Yes

4. Is the manuscript presented in an intelligible fashion and written in standard English?

Reviewer #1: Yes

Reviewer #2: Yes

5. Review Comments to the Author

Reviewer #1: In this manuscript, the melanin synthesis pathway key genes ebony and tan were identified to regulate the pigmentation and body color patterns in the cricket Gryllus bimaculatus, as a model of the hemimetamorphosis insects. Authors found that dynamic alteration in expression levels of Gb’ebony and Gb’tan in combination was correlated with the body color transition from nymphal stages through adult and knockout of Gb’ebony and Gb’tan gene resulted darkened systemically and displayed a yellow color in certain areas and stages, respectively. ebony and tan, as some conservative functional genes, have been studied in many insect species including Drosophila melanogaster, Tribolium casteneum, Henosepilachna vigintioctopunctata, Oncopeltus fasciatus, Bombyx mori, Spodoptera litura and so on. This paper extends the understanding of ebony and tan to regulate the body color pattern through nymph to adult stage in the hemimetamorphosis insect. However, some critical points should be addressed prior to further its consideration for publication.

Major comments:

1.According to the results of Figure 3, authors showed the expression profiles of Gb’ebony and Gb’tan transcripts from embryo through adult stages. There were some unexpressed points, such as the D3 of 5th, the D3 of 7th, the D3 of 8th and D3 of adult stage. Please display the time and tissue of the detected samples. And it is better for the expression profiles of genes using the means labeled with different letters indicate significant difference at P < 0.05. Two segments of Y axis are more visual to show the low expressed stage in Fig 3A and 3A’.

2.In Figure 4, authors demonstrated that the knockout of Gb’ebony and Gb’tan genes were successful and the results were solid. But there absented a wild type control of Fig 4A. Authors should display the detail information of the enzyme (Gluide-it Resolvase) including company, place of production, experimental methods and so on. A PCR sequencing of the G0 eggs is better to show the mutant results.

3.From Figure 5 and Figure 6, authors showed many phenotypes of Gb’ebony and Gb’tan mutants. According to previous reports, the functions of gene ebony and tan are opposite. The phenotypes of Gb’ebony mutants were obvious and convinced. What is different about the homozygous mutants Gb’ebonycr1 and Gb’ebonycr2 ? Why authors chose the homozygous mutants Gb’ebonycr1 to show the mutant phenotypes? Some other phenotypes could display in the supplemental material. The homozygous mutants Gb’tan showed different phenotypes in 7th and 8th. Please explain the related and potential reasons. The western blot of Gb’tan gene is not clear and enough to identify the gene function loss of mutant strain of Gb’tan. qRT-PCR is a full complement of these results.

4.The sexual dimorphism is ubiquitous in animals and plants, especially in insects. The body color and pigmentation are the typical trait in insect sexual dimorphism. Authors found there are some different phenotypes of body color between male and female adult in G. bimaculatus. The hormones and sex determination genes could regulate the expression of Gb’ebony and Gb’tan. It is an interesting point for researchers.

Others:

1.Line 262, 263 and 398, Gb'ebony should be Gb’ebony.

2.The format of scale bars should be unified in Figure 5 and 6.

3.It is better to upload the Gb’ebony and Gb’tan sequences to NCBI and get the GenBank access number.

Reviewer #2: The study named "Combinatorial expression of ebony and tan generates body color variation from nymph through adult stages in the cricket, Gryllus bimaculatus" nicely addressed the functional of melnin related genes ebony and tan in cricket, the understanding could enrich the konwledge of insect melnin and even the technology used in this study conuld extend the potential of the other insects.

One critical issue, the potential off target effects should be included in the results.

6. PLOS authors have the option to publish the peer review history of their article (what does this mean?). If published, this will include your full peer review and any attached files.

Reviewer #1: No

Reviewer #2: No

---

## [Author Response · Author response to Decision Letter 0]

4 May 2023

Response to reviewer comments

We would like to thank the reviewers for their careful and insightful review of our manuscript. We have considered all points raised and provide point-by-point responses below.

Reviewer #1

Comment1:

According to the results of Figure 3, authors showed the expression profiles of Gb’ebony and Gb’tan transcripts from embryo through adult stages. There were some unexpressed points, such as the D3 of 5th, the D3 of 7th, the D3 of 8th and D3 of adult stage. Please display the time and tissue of the detected samples. And it is better for the expression profiles of genes using the means labeled with different letters indicate significant difference at P < 0.05. Two segments of Y axis are more visual to show the low expressed stage in Fig 3A and 3A’. 

Response:

The time and tissue of the detected samples

We did not adequately describe the sampling conditions for crickets. Therefore, the following text was added to the ‘RT-qPCR’ section of the Methods:

Lines 474–477: “All crickets were sampled during the dark period. Unpigmented and pigmented crickets on day 1 of each stage were sampled within 1 h and between 2–15 h, respectively, after hatching and molting. mRNA extraction and reverse transcription were performed from embryos, nymphs, and adult whole bodies according to the method described above (the cDNA cloning section).”

In addition, the text related to this has been revised in the Results and figure caption.

 Modification of Fig 3

Our explanation may have been misleading. We previously used asterisks on day 1 of each stage at the first to third instar to denote significant differences in gene expression levels in unpigmented and pigmented crickets during the same day. Therefore, we have circled in red the areas of the graph that show rapid changes in the expression levels on day 1 of each stage at the first to third instar and added the results of the significance tests. Asterisks at the seventh instar to adult stage also indicated significant differences in expression levels in females and males at the same stage. 

Following the reviewer’s advice, Fig. 3A’ was deleted, and a new graph was created as Fig. 3A, with the Y-axis divided into two parts. In this revised figure, the X-axis of the graph and the plot were changed to reflect the actual timescale. The scale on the X-axis indicates one day. We have corrected the parts of the Results and figure captions that correspond to these changes.

Comment2:

In Figure 4, authors demonstrated that the knockout of Gb’ebony and Gb’tan genes were successful and the results were solid. But there absented a wild type control of Fig 4A. Authors should display the detail information of the enzyme (Guide-it Resolvase) including company, place of production, experimental methods and so on. A PCR sequencing of the G0 eggs is better to show the mutant results. 

Response:

A wild type control of Fig 4A

Data from negative control experiments with wild-type eggs have been added to Fig 4A. The data before trimming was placed in the supporting information file (S1_raw_images). The following text was added to the Results: 

Lines 191–194: “PCR products from the wild-type were not cleaved by the endonuclease. Raw data from this experiment can be found in the supporting information file (S1_raw_images).”

The detail information of the enzyme (Guide-it Resolvase)

Detailed information on the enzyme (Guide-it Resolvase) was added to the Results and ‘Genotyping’ section of the Methods as shown below (the underlined parts are the revised sections):

Lines 190 (Results): “Guide-it Resolvase (Takara Bio, Shiga, Japan).”

Lines 465–470 (Methods): “Mutations at the gRNA targeted-regions of Gb’ebony and Gb’tan in the G0 and G1 embryos were detected by amplifying these regions and digesting the PCR products with the Guide-it Mutation Detection Kit (Takara Bio, Shiga, Japan) based on a previously reported method [13, 30]. Briefly, the PCR products were reannealed under the following conditions: 95℃ for 5 min, 95℃–85℃ at -2℃/sec, and 85℃–25℃ at -0.1℃/sec. Next, 10 μL of the PCR products were mixed with 5 μL of distilled water and 1 μL of Guide-it Resolvase (Takara Bio) in the Guide-it Mutation Detection Kit (Takara Bio). This was incubated at 37℃ for 15 min. Then, mutagenesis in the gRNA-targeted region was detected by electrophoresis using 3% agarose gel.”

A PCR sequencing of the G0 eggs

PCR sequencing data for G0 eggs has been added to the supporting information as S2 Fig. The following text was added to the Results and caption:

Lines 193–194 (Results): “Furthermore, sequencing analysis confirmed the introduction of mutations in the crRNA target region of G0 eggs (S2 Fig).”

Lines 605–607 (in S3 Fig caption): “S3 Fig. PCR sequencing of G0 eggs

Sequence analysis of mutations introduced around the crRNA target region (highlighted in yellow) in the Gb’ebonycr1 (A), Gb’ebonycr2 (B), and Gb’tancr1 (C) mutants of the G0 generation.” 

Comment3: 

From Figure 5 and Figure 6, authors showed many phenotypes of Gb’ebony and Gb’tan mutants. According to previous reports, the functions of gene ebony and tan are opposite. The phenotypes of Gb’ebony mutants were obvious and convinced. What is different about the homozygous mutants Gb’ebonycr1 and Gb’ebonycr2 ? Why authors chose the homozygous mutants Gb’ebonycr1 to show the mutant phenotypes? Some other phenotypes could display in the supplemental material. The homozygous mutants Gb’tan showed different phenotypes in 7th and 8th. Please explain the related and potential reasons. The western blot of Gb’tan gene is not clear and enough to identify the gene function loss of mutant strain of Gb’tan. qRT-PCR is a full complement of these results.

Response:

Phenotypes of Gb’ebony

As shown in Fig. 4B, the two strains of Gb'ebony mutants, Gb’ebonycr1 and Gb’ebonycr2, differ in the crRNA used for genome editing and in the sequence of the introduced mutations. There was no phenotypic difference between these mutants. As evidence, a photograph of one mutant, Gb’ebonycr2, was added to the supplemental material as S5 Figure. We added the following text to the Results and caption:

Lines 266–270 (Results): “We also analyzed the Gb'ebonycr2 strain, which was generated using a different gRNA (crRNA2) than that used in the generation of the Gb'ebonycr1 strain. The Gb'ebonycr2 mutants exhibited systemic darkening of body color at all stages (S5 Fig), indicating that there were no phenotypic differences between the two mutant strains. This result also indicates that there were no off-target effects on the phenotype.”

Lines 615–620 (in S5 Figure caption): “S5 Fig. Phenotype of the Gb’ebonycr2 mutant strain. (A) Dorsal views of wild-type and Gb’ebonycr2 mutant adults. (B) Effect of Gb’ebony knockout on the color of adult wings. FW: Forewing, HW: Hind wing. (C) Dorsal views of wild-type and Gb’ebonycr2 mutant nymph stages. (C’) Magnified image of the dorsal side of the thorax and the tail in first instar nymphs. See Figure 5 for a picture of the wild-type. Scale bars: 10 mm in A and B; 2 mm (1st–6th instar nymphs) and 10 mm (7th–8th instar nymphs) in C; 0.5 mm in C’.”

Phenotypes of Gb’tan

As noted in the results, a quantitative comparison of body color was performed at 7th instar, when the body color change was observed, and showed a significant difference relative to the wild type. On the other hand, we did not obtain a clear phenotype in the other stages, including the 8th instar (the original photos of the 8th instar of the Gb’tan mutant in Fig. 6C was misleading and has been replaced). We are currently considering the following reasons for the phenotype, especially at 7th instar:

 As there is no marked difference in the expression level of Gb'tan between stages in the late instar, it is unlikely that it is responsible for the seventh instar-specific body color change. Tan is an enzyme that synthesizes dopamine using NBAD as a substrate, and the extent of its contribution in vivo is thought to depend not only on its own expression level but also on the intracellular amount of NBAD and the expression of the NBAD synthase Ebony. At the seventh instar stage, Gb’ebony expression has reached peak levels, implying that NBAD levels may have peaked as well. This means that the contribution of Gb’tan is higher at this stage, and a knockout phenotype has emerged. This hypothesis is also supported by data showing that wild-type males express more Gb’ebony in seventh instar nymphs and adults (Fig 3A), and males of Gb’tan mutants at this stage had a more clearly observable phenotype than females (Fig 6). 

The text of the discussion related to these points has been revised (Lines 368-381).

Identifying the gene function loss of mutant strain of Gb’tan.

We compared the amount of Gb'tan transcripts in the wild-type and Gb'tan mutant strains by RT-qPCR and added the results to Figure 4D. The results showed that the expression of Gb'tan in the Gb'tan mutant strain was significantly lower than that in the wild-type. We added the following text to the Results and caption:

Lines 233–237 (Results): “To further validate the Gb'tan knockout, the transcript levels of Gb'tan were analyzed by RT-qPCR. The results showed that the amount of Gb'tan transcripts in the Gb'tan mutant line was significantly reduced to about one-tenth of that in the wild-type (P < 0.01, Fig 4D), demonstrating that mutagenesis was effective at the transcriptional level.”

Lines 204–208 (in Figure 4 caption): “(D) Effects of Gb'tan knockout on Gb'tan transcription. RNA was extracted from the whole body of day 1 seventh instar nymphs of the wild-type and Gb'tan mutants within 1 h of molting and subjected to RT-qPCR. Data are presented as the mean value ± SD obtained from four biological replicates and three technical replicates. The asterisks (**) mean statistical significance at P < 0.01 in a Student’s t-test.”

Comment4: 

The sexual dimorphism is ubiquitous in animals and plants, especially in insects. The body color and pigmentation are the typical trait in insect sexual dimorphism. Authors found there are some different phenotypes of body color between male and female adult in G. bimaculatus. The hormones and sex determination genes could regulate the expression of Gb’ebony and Gb’tan. It is an interesting point for researchers. 

Response:

Thank you for this comment. We agree that Gb’ebony and Gb’tan could be regulated by hormones and sex determination genes. As a next step, we plan to study the regulatory mechanisms of the expression of these genes

Reviewer #2 comment:

The study named "Combinatorial expression of ebony and tan generates body color variation from nymph through adult stages in the cricket, Gryllus bimaculatus" nicely addressed the functional of melanin related genes ebony and tan in cricket, the understanding could enrich the knowledge of insect melanin and even the technology used in this study could extend the potential of the other insects. One critical issue, the potential off target effects should be included in the results.

Response: 

Following this comment, we sequenced the potential off-target sites of the crRNA used to generate the Gb'tan mutant, as only one mutant strain was generated in the study. As a result, no mutations were identified as potential off-target sites. This result was added to the S3 Figure. Additionally, we added the following text to the Results and caption:

Line 215–216 (Results): “Only one Gb'tan mutant strain was generated in this study, but mutations in off-target sites in Gb’tan crRNA1 (S1 Fig) were not observed in sequence analysis (S3 Fig).”

Lines 608–614 (figure caption): “S4 Fig. Sequence analysis of the potential off-target sites of crRNAs used to generate Gb'tan mutant strains. PCR amplification and sequence analysis of off-target sites (highlighted in blue) predicted for Gb'tan crRNA1 ((A) Scaffold307 (GenBank: BOPP01000307.1), (B) Scaffold70 (GenBank: BOPP01000070.1), and (C) Scaffold18 (GenBank: BOPP01000018.1) on G. bimaculatus genomic data (GenBank: GCA_017312745.1)) were performed using the genome of the Gb'tan mutant as a template and the primers listed in S1 Table compared with the wild-type.” 

Others

Comment1: Line 262, 263 and 398, Gb'ebony should be Gb’ebony. 

Response: We have corrected this.

Comment2: The format of scale bars should be unified in Figure 5 and 6.

Response: We have made this change.

Comment3: It is better to upload the Gb’ebony and Gb’tan sequences to NCBI and get the GenBank access number.

Response:

As mentioned in the text, we registered their sequence information with the DNA Data Bank of Japan (DDBJ). When sequence information is made public, it is automatically shared with the NCBI. Currently, the sequence information is scheduled to be released on 2023-05-01. If this paper is published earlier than that, we will release this information immediately. The DDBJ guidelines are published online at https://www.ddbj.nig.ac.jp/about/index-e.html as follows:

“In Japan, DDBJ Center internationally contributes as a member of INSDC to collect and to provide nucleotide sequence data with ENA/EBI in Europe and NCBI in USA. DDBJ Center is officially certified to collect nucleotide sequences from researchers and to issue the internationally recognized accession number to data submitters. The accession number issued for each sequence data is unique on the database and internationally recognized to guarantee the submitter the property of the submitted and published data. Since DDBJ Center exchanges the released data with ENA/EBI and NCBI on a daily basis, the three data centers share virtually the same data at any given time. The virtually unified database is called INSD; International Nucleotide Sequence Database. DDBJ collects sequence data mainly from Japanese researchers, but of course accepts data and issue the accession numbers to researchers in any other countries. 99% of INSD data from Japanese researchers are submitted through DDBJ.”

---

## [Editor Report · Decision Letter 1]

5 May 2023

Combinatorial expression of *ebony* and *tan* generates body color variation from nymph through adult stages in the cricket, *Gryllus bimaculatus*

PONE-D-23-01091R1

Dear Dr. Mito,

We’re pleased to inform you that your manuscript has been judged scientifically suitable for publication and will be formally accepted for publication once it meets all outstanding technical requirements.

Kind regards,

Peng He, Ph.D

Academic Editor

PLOS ONE
---

## [Editor Report · Acceptance letter]

9 May 2023

PONE-D-23-01091R1 

Combinatorial expression of *ebony* and *tan* generates body color variation from nymph through adult stages in the cricket, *Gryllus bimaculatus*

Dear Dr. Mito:

I'm pleased to inform you that your manuscript has been deemed suitable for publication in PLOS ONE. Congratulations! Your manuscript is now with our production department. 

Kind regards, 

on behalf of

Dr. Peng He 

Academic Editor

PLOS ONE